# Development of a novel humanized gut-brain axis model as a tool toward personalized nutrition
Myrto S. Chatzopoulou [1], Ravi Vumma [2,3], Samira Prado [1,4], Mathias W. Scharf [1],
Victor Castro-Alves [4], Ashley N. Hutchinson[1], Ignacio Rangel[1], Tatiana M. Marques[1], Rebecca Wall[1],
Robert J. Brummer [1] & Julia Rode [1,5] ✉

Intestinal luminal microbial metabolites affect tryptophan and serotonin metabolism, and cross or modify the blood-brain barrier (BBB). Understanding those mechanisms further necessitates integrated gut-brain axis model systems. Using an ex vivo-in vitro approach, $H_2O_2$-stressed or non-stressed human dermal fibroblasts – representing the BBB – are cultured with serosal fluids of healthy or irritable bowel syndrome human colonic biopsies collected from Ussing chamber experiments, after participant's colon was exposed to butyrate in vivo, fecal fiber fermentation or control supernatant ex vivo. Culturing fibroblasts with serosal fluids does not compromise viability or have cytotoxic effects. Serosal fluids alone do not alter expression of tryptophan-related large amino acid membrane transporter genes and proteins, nor their activity (i.e., tryptophan uptake). However, adding serosal fluids to fibroblasts prior to oxidative stress indicate a protective role. This new model allows investigation of direct effects of serosal content on BBB-representing fibroblasts and is highly promising for more personalized applications.

The role of intestinal microbes in the regulation of the immune system, metabolism of nutrients, and production of neuroactive compounds has established them as key modulators of the gut-brain axis[1–4], the homeostatic, reciprocal communication between the human intestine and the brain[2,5–7]. Disorders of gut-brain interactions, like irritable bowel syndrome (IBS), are characterized by low-grade inflammation and altered bowel habits, but they are also often associated with altered central nervous system (CNS) processing of gut sensory signals, as well as anxiety and depression[8–10]. Although growing evidence has suggested that the gut microbiota impact brain function and behavior, mainly in the context of neurodegenerative and psychiatric disorders, the potential underlying molecular and cellular mechanisms remain vastly uncovered[2,4,11–14].

The translocation of certain gut microbiota-derived compounds from the luminal/mucosal side to the basolateral/serosal side of the intestine, and subsequently into the bloodstream, can affect gut-brain signaling. This translocation influences CNS function by modulating systemic levels of neuroactive compounds that can eventually cross or modulate the permeability of the blood-brain barrier (BBB)[2,12,15,16].

Gut microbiota may influence microbial and human metabolism, such as of tryptophan[17], which is utilized for the biosynthesis of 5-hydroxytryptamine (5-HT), also known as serotonin. Although 95% of the total 5-HT is synthesized by enterochromaffin cells in the gut, gut microbiota can also moderate or contribute to that synthesis[18–22]. Intestinal 5-HT is unable to pass through the BBB; however, it can bind to 5-HT$_3$ receptors on afferent vagal neurons locally and ultimately affect stress response and behavior in the CNS[17,23–25].

Interestingly, increasing evidence suggests that a high-fiber diet can have suppressive effects on microbial tryptophan degradation, allowing levels of circulating tryptophan and consequently central 5-HT to rise[18,23]. One reason for this could be the higher luminal availability of short-chain fatty acids (SCFAs) deriving from the fermentation of indigestible dietary fiber by the microbiota in the colon. Butyrate is one of the mainly produced SCFAs that is rapidly absorbed by the gut mucosa and provides an excellent energy source for colonocytes[26]. In addition, microbial synthesis of SCFAs, especially butyrate, has been associated with the suppression of the kynurenine pathway, which ameliorates levels of systemic tryptophan[27,28].

[1]School of Medical Sciences, Faculty of Medicine and Health, Örebro University, 70182 Örebro, Sweden. [2]Department of Chemistry and Biomedical Sciences, Linnaeus University, 39231 Kalmar, Sweden. [3]Department of Pharmaceutical Sciences and Administration, School of Pharmacy, University of New England, Portland, 04103 ME, USA. [4]School of Science and Technology, Faculty of Science, Business and Engineering, Örebro University, 70182 Örebro, Sweden. [5]School of Health Sciences, Faculty of Medicine and Health, Örebro University, 70182 Örebro, Sweden. ✉e-mail: Julia.rode@oru.se

Moreover, endogenous butyrate enters the bloodstream and infiltrates the BBB in low concentrations, potentially exerting positive neurological effects[1,29], while exogenous butyrate at pharmacological doses has been shown to generate prominent effects on both the brain and BBB in various animal studies[1,30,31]. For instance, increased expression of tight junctions and decreased permeability of the BBB were observed after mono-colonization of germ-free mice with butyrate-producing bacteria or by oral administration of sodium butyrate[30]. Even though animal models provide a strong foundation to study the impact of gut microbiota alterations on the BBB, they inevitably fail to capture certain human physiological aspects (e.g., brain function, immune system, gut microbiota composition).

This weakness, along with sustainability and animal welfare concerns, has encouraged the rapid advancement of in vitro models which can significantly reduce or even deplete the need for animal studies[32]. In vitro modeling with dynamic cell culture conditions has been considered one of the most promising approaches to mimic the microbiota-gut-brain axis' complexity[32,33]. In vitro BBB models are mainly based on endothelial cell cultures or co-cultures with other perivascular cells (i.e., pericytes, and astrocytes) in two- or three-dimensional platforms[34,35]. Moreover, single- and multi-organ platforms, based respectively on the "organ-" or "body-on-a-chip" concept, have also been proposed[36–39], while more and more intestinal and BBB models strive to implement the use of induced pluripotent stem cells[34,37,40–44].

Following these rapid developments, our model uses a combination of ex vivo and in vitro experimental techniques to explore the potential interactions of gut-derived metabolites and cells representing the endothelium of the BBB.

In this study, we hypothesized that microbial metabolites, such as SCFAs, after passage through intestinal tissue, affect the uptake of serotonin-precursor tryptophan in fibroblasts.

## Results

### Characterization of serosal fluids—microbiology, pH, protein and cytokine content, metabolite profile

Serosal fluids showed no microbial growth of aerobic bacteria, had a pH of ~8.5 and contained very low protein concentrations (below the detection limit of 0.1 mg/mL). However, all ten proinflammatory cytokines—interleukin (IL)-2, IL-4, IL-6, IL-8, IL-10, IL-12p70, IL-13, interferon gamma (IFN-γ), and tumor necrosis factor alpha (TNF-α)—were detected in low levels with the more sensitive MSD® immunoassay in the different serosal fluids, and G-Krebs solution (Supplementary Fig. 1).

Figure 1 shows the relative concentration (μM) of six different compounds of short-chain fatty acid (SCFA) metabolism in the different types of serosal fluids, based on internal standards with known concentrations. Serosal fluids deriving from fecal fermentation supernatant-exposed biopsies had visually higher concentrations of caproate, propionate, and valerate, but lower concentrations of acetate compared with biopsies sampled after in vivo butyrate infusion. Butyrate and isovalerate concentrations seemed visually similar among all serosal fluids, and the total concentration of SCFAs was the highest in the HC-FERFIB sample. Nevertheless, statistical comparisons were not feasible since metabolite profiles of pooled serosal fluids were measured once (n = 1).

Measurements of SCFAs, as well as other metabolites involved in the tricarboxylic acid cycle (TCA) and tryptophan metabolism, are also presented as log2 fold changes against the serosal carrier solution G-Krebs (Supplementary Fig. 2). Propionate and caproate were visually increased in the ex vivo manipulation serosal fluids (HC-FERFIB/HC-FERCON) compared to G-Krebs, whereas propionate was increased, but caproate decreased in the in vivo manipulations-derived serosal fluids (HC-BUT/HC-CON, IBS-BUT/IBS-CON). Interestingly, all TCA metabolites were visually decreased in the HC-FERFIB serosal fluid, while fold changes of tryptophan-related metabolites varied across the different serosal fluids but were overall visually more increased in the HC-CON group.

### Exposure of fibroblasts with serosal fluids does not affect morphology, cell viability, cytotoxicity

Microscopic examination of the cell cultures revealed typical spindle-shaped adhesive fibroblasts across all conditions.

Acute (1 h) and relatively extended (24 h) exposure of fibroblasts with serosal fluids did not have any effects on cytotoxicity (measured as cell membrane integrity) nor cell viability (measured as metabolic activity) compared to serosal fluids' carrier solution, i.e., physiological G-Krebs solution (Supplementary Figs. 3 and 4).

Exposure of healthy cells to 10 μM $H_2O_2$ evoked stress, visually hampering metabolic activity, without being cytotoxic to the cells (Supplementary Fig. 5). Neither the exposure of serosal fluids as a preventive measure (before oxidatively stressing the cells), nor as a treatment measure (after oxidatively stressing the cells) resulted in significant alterations of viability or cytotoxicity as compared to incubation with G-Krebs pre and post oxidative stress (Supplementary Figs. 3 and 4).

### Effect of serosal fluids on tryptophan transporters—gene expression, protein expression, and protein activity

Gene expression of SLC7A5, SLC7A8, and SLC3A2. For acute (1 h) and relatively extended (24 h) exposure with serosal fluids alone, all conditions showed similar gene expression levels of the tryptophan transporters. The LAT1-specific subunit SLC7A5 and LAT2-specific subunit SLC7A8, as well as the shared subunit SLC3A2, were not significantly affected by serosal fluids compared to G-Krebs.

The preventive measure of 24 h exposure with serosal fluids before oxidatively stressing (10 μM $H_2O_2$) the cells significantly altered gene expression of SLC7A5 and SLC7A8 (p < 0.05), while the expression of subunit SLC3A2 was not significantly affected (Supplementary Fig. 6). Namely, the expression of SLC7A5 showed a significant, and biologically relevant (i.e., below the 0.5-fold mark, typically considered as cutoff for biologically relevant downregulation (vs above 2-fold increase is typically considered a biologically relevant upregulation)), decrease in all conditions compared to G-Krebs (HC-CON: -9.1 ± 2.9, p = 0.0052; HC-BUT: -8.9 ± 2.9, p = 0.0063; IBS-CON: -8.7 ± 2.7, p = 0.0040; IBS-BUT: -8.4 ± 2.7, p = 0.0053; HC-FERFIB: -9.5 ± 2.7, p = 0.0022; all F(7, 19) = 3.1) (Fig. 2). The expression of SLC7A8 was significantly decreased after the exposure of fibroblasts with serosal fluids from healthy subjects and IBS patients (HC-CON and IBS-CON) and those from fecal fiber fermentation supernatant (HC-FERFIB), though only the exposure with HC-FERCON exceeded the cut-off for biological relevance (below the 0.5-fold mark) (Fig. 3).

For the treatment measure of 24 h incubation with serosal fluids after oxidatively stressing (10 μM $H_2O_2$) the cells, the expression of SLC7A8 was biologically relevant (below the 0.5-fold mark) and significantly decreased for the HC-CON condition (-0.3 ± 0.2, p = 0.0477, F(7, 14) = 5.1) and non-significantly (p = 0.0518) for the HC-FERFIB condition, compared to G-Krebs (Fig. 3); on the other hand, the expression of SLC7A5 and SLC3A2 was unaffected (Fig. 2 and Supplementary Fig. 6).

Gene expression for the preventive and treatment conditions of acute (1 h) exposure was not assessed due to insufficient quality of extracted RNA.

Protein expression of LAT1 and LAT2. The expression of the LAT1 protein was not significantly affected by any of the tested conditions— sole serosal fluid exposure, preventive measure nor treatment measure— in neither acute (1 h) nor relatively extended (24 h) exposures (Fig. 4).

The expression of LAT2 was not significantly affected by acute (1 h) or relatively extended (24 h) exposure to serosal fluids alone (Fig. 5). However, LAT2 expression was affected when serosal fluids were applied in combination with oxidative stress. While the treatment exposure with serosal fluids for 24 h revealed no effects, exposure to serosal fluids exposed to fecal fiber fermentation supernatants (HC-FERFIB) for 1 h significantly increased expression of LAT2 ( + 0.2 ± 0.1, p = 0.137, F(7, 15) = 3.5) compared to G-Krebs.

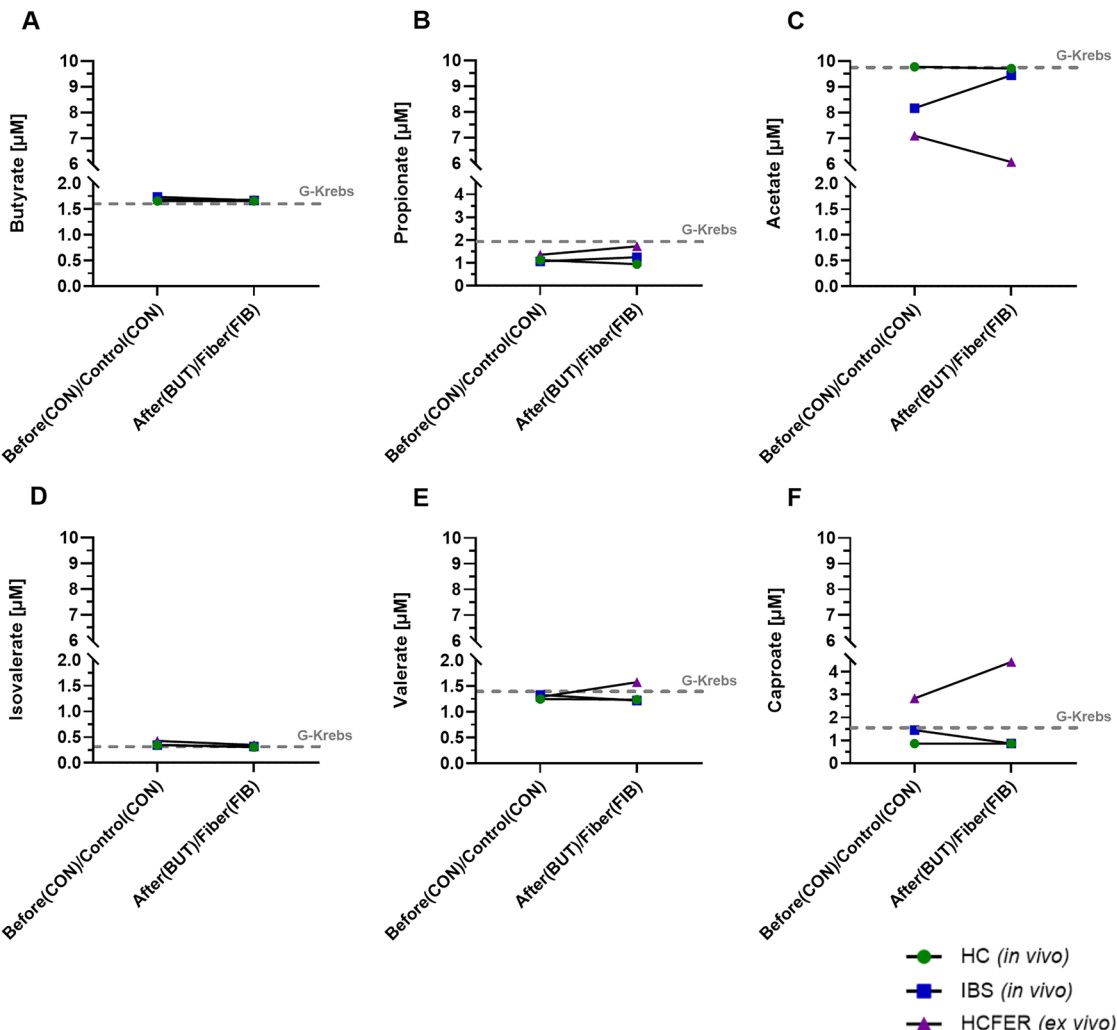

**Fig. 1 | Concentration [µM] of six short-chain fatty acid (SCFA)-related metabolites in serosal fluids quantified based on internal standards. A** Butyrate, (**B**) Propionate, (**C**) Acetate, (**D**) Isovalerate (3-Methylbutanoic acid), (**E**) Valerate, and (**F**) Caproate. Presented from left to right are the control exposures as "Before", meaning *before* in vivo butyrate infusion (HC-CON/IBS-CON), or "Control", meaning ex vivo exposure with the control fecal fermentation supernatant (HC-FERCON); and the treatment exposures as "After" for after butyrate infusion (HC-

BUT/IBS-BUT), or "Fiber" for the exposure to the fecal fermentation supernatant with fiber (HC-FERFIB). The gray dashed line represents the concentration measured in the physiological serosal carrier-solution (G-Krebs). Green circles present HC-CON and HC-BUT, blue squares present IBS-CON and IBS-BUT, purple triangle presents HC-FERCON and HC-FERFIB. Metabolomics of pooled serosal fluids were assessed without replicates, hence every datapoint presents a single measure.

**Protein activity.** For acute (1 h) and relatively extended (24 h) exposure with serosal fluids alone, tryptophan uptake was not significantly altered (Fig. 6).

Exposure of otherwise healthy cells to 10 µM $H_2O_2$ evoked stress, decreasing tryptophan uptake by >10% when exposed to G-Krebs for 1 h before or after the stressor (Supplementary Fig. 7), which is comparable to our previous work[45].

For the 24 h, but not 1 h, preventive condition (hence before oxidatively stressing the cells), tryptophan uptake was significantly increased after incubation with HC-FERFIB ($+ 6.9 \pm 1.9$ nmol/mg protein, $p = 0.0015$, F(7, 22) = 2.7) compared to G-Krebs, while the treatment conditions did not evoke any effects (Fig. 6).

**Molecular consequences—metabolite profile of cell harvest and conditioned medium.** Results of the metabolomics analysis with focus on polar metabolites with special interest in amino acids and tryptophan pathway are presented in Supplementary Figs. 8 and 9. It can be observed that cell harvests have a distinct metabolite profile compared to the supernatants, with less metabolites being detected in the former. The

patterns of fold changes between metabolite profiles between the acute (1 h) and relatively extended (24 h) conditions seem visually similar, with the fold changes intensifying for certain metabolites for the 24 h exposure to serosal fluids.

## Discussion

Following the advancement of gut-brain axis models, this study aimed to establish an ex vivo-in vitro model for the passage of luminal metabolites through the gut barrier and their subsequent impact on the blood-brain barrier (BBB) through the humoral route, applicable for precision nutrition assessments. By collecting serosal fluids of distal colonic biopsies after Ussing chamber experiments, we attempted to imitate the passage of microbial metabolites, such as short-chain fatty acids (SCFAs) (i.e., acetate, propionate, butyrate), from the luminal to the serosal side of the gut lining and to partially capture the secretome of the biopsy itself.

To focus on the humoral route of gut-brain communication and to test how these fluids and their content could impact the BBB, we introduced them in the culture media of a human dermal fibroblast cell line. Fibroblasts present similarities in the expression of amino acid transport systems, such

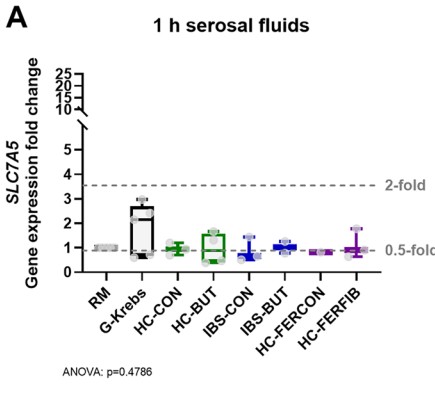

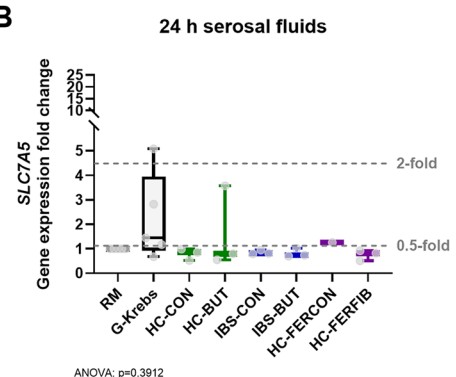

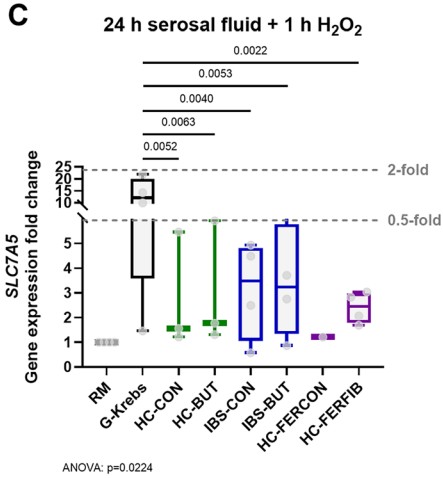

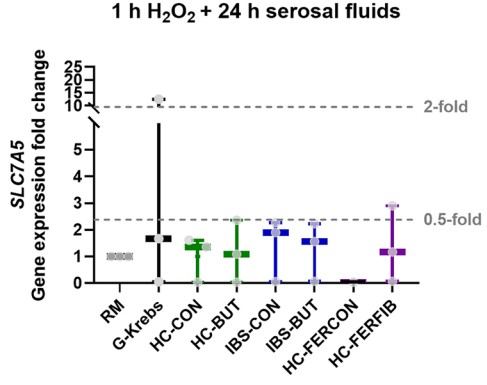

**Fig. 2 | Gene expression of subunit *SLC7A5* of the major tryptophan transport LAT1 normalized to reference gene *GAPDH* and the methodological reference condition regular medium. A** 1 h exposure to serosal fluids alone, (**B**) 24 h exposure to serosal fluids alone, (**C**) exposure to 24 h serosal fluids as preventive measure before oxidatively stressing the cells (by 10 μM $H_2O_2$), and (**D**) exposure to 24 h serosal fluids as a treatment measure after oxidatively stressing the cells. *X*-axis presents exposure to regular medium (RM), serosal fluid carrier solution G-Krebs, and serosal fluid pool derived from healthy biopsies collected from the unexposed colon (HC-CON), from healthy biopsies collected after in vivo butyrate exposure (HC-BUT), from irritable bowel syndrome biopsies collected from the unexposed colon (IBS-CON), from IBS biopsies collected after in vivo butyrate exposure (IBS-BUT), from healthy biopsies ex vivo exposed to supernatant of fecal control

fermentation (HC-FERCON), and from healthy biopsies ex vivo exposed to supernatant of fecal fiber fermentation (HC-FERFIB). Per condition n ≤ 5 biologically independent experiments. Data visualized as boxplots, horizontal line indicates median, whiskers indicate minimum to maximum, individual data points are overlayed. Black boxpots present control conditions RM and G-Krebs, green presents HC-CON and HC-BUT, blue presents IBS-CON and IBS-BUT, purple presents HC-FERCON and HC-FERFIB. Commonly used thresholds for biological relevance indicated by the gray dashed lines for 0.5- and 2-fold changes compared to G-Krebs. One-way ANOVA with posthoc uncorrected Fisher's LSD multiple comparisons test versus G-Krebs control condition. Exact *p*-value for ANOVA and if significant for posthoc tests provided in figure.

as tryptophan transporters LAT1 and 2, with brain microvascular endothelial cells that comprise the BBB[46,47] and are often used to model psychiatric[48,49] and neurodegenerative disorders[50–52].

Additionally, we applied our model to test whether serosal fluids could differentially impact the BBB by using proxies of fiber-rich diets (in vivo butyrate exposure, ex vivo exposure to supernatant of fecal fermentation enriched with wheat-bran fiber fractions) from a healthy or irritable bowel syndrome (IBS) intestinal environment. Lastly, we implemented the use of $H_2O_2$ as a stressor, thus partly assessing potential therapeutic or protective effects of the serosal fluids.

Serosal fluids had a close to physiological pH and did not present any microbial growth of aerobic bacteria, although the Ussing chamber is an open non-sterile system. As per protocol, fibroblasts were cultured with addition of antibiotics to the culture medium and at no point microbial contaminations were observed. After acute (1 h), as well as relatively extended (24 h) exposure to serosal fluids, fibroblasts showed typical spindle-shaped morphology and typical adhesive characteristics examined by microscopic examination. Two molecular biological tests showed that the

cells were normally, metabolically active (cell viability assays) and had a normal cell membrane integrity (cytotoxicity assays). Hence, those findings support the feasibility of such experiments, and applied concentrations can be considered as adequate.

For both acute and relatively extended exposure solely to serosal fluids, neither gene expression, nor protein expression, nor protein activity (*i.e.*, transmembrane tryptophan transport) were significantly affected.

To evoke oxidative stress, the usage of $H_2O_2$ in vitro is common practice, and we have validated that the stressor worked in our setting by compromising the tryptophan transporter system (see Supplementary Fig. 7). However, tryptophan transport remained unchanged and in one case (HC-FERFIB) even increased when the cells were exposed to serosal fluids prior to oxidative stress, which could indicate a protective effect of the serosal fluids against oxidative damage effects, further supported by the alterations we see in gene and protein expression levels.

Interestingly, as a preventive measure for stressed cells, all serosal fluids, especially HC-FERFIB, significantly decreased gene expression of relevant tryptophan transporters. Despite the pronounced effect on

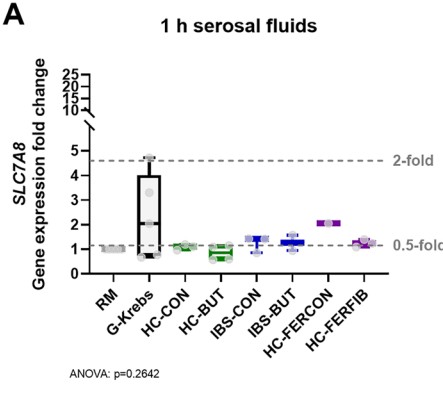

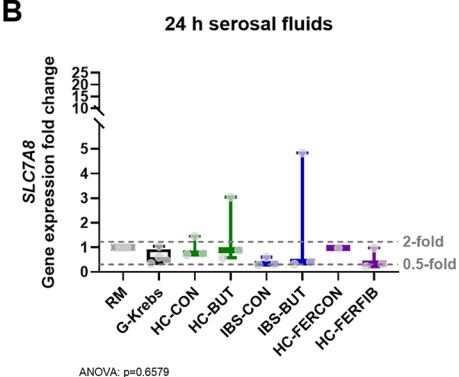

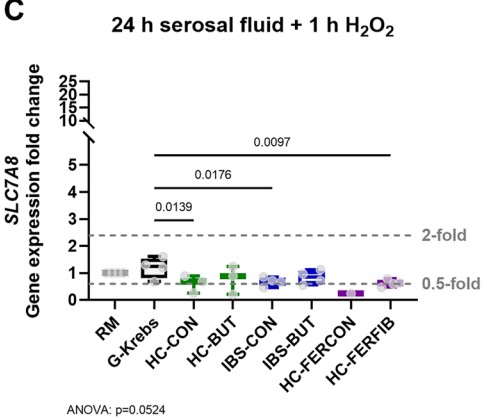

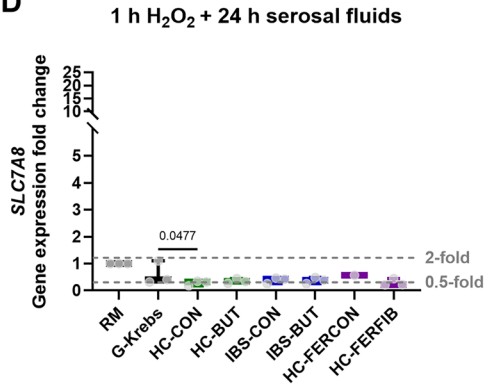

**Fig. 3 | Gene expression of subunit *SLC7A8* of the minor tryptophan transport LAT2 normalized to reference gene *GAPDH* and the methodological reference condition regular medium. A** 1 h exposure to serosal fluids alone, (**B**) 24 h exposure to serosal fluids alone, (**C**) exposure to 24 h serosal fluids as preventive measure before oxidatively stressing the cells (by 10 μM $H_2O_2$), and (**D**) exposure to 24 h serosal fluids as a treatment measure after oxidatively stressing the cells. *X*-axis presents exposure to regular medium (RM), serosal fluid carrier solution G-Krebs, and serosal fluid pool derived from healthy biopsies collected from the unexposed colon (HC-CON), from healthy biopsies collected after in vivo butyrate exposure (HC-BUT), from irritable bowel syndrome biopsies collected from the unexposed colon (IBS-CON), from IBS biopsies collected after in vivo butyrate exposure (IBS-BUT), from healthy biopsies ex vivo exposed to supernatant of fecal control

fermentation (HC-FERCON), and from healthy biopsies ex vivo exposed to supernatant of fecal fiber fermentation (HC-FERFIB). Per condition $n \le 5$ biologically independent experiments. Data visualized as boxplots, horizontal line indicates median, whiskers indicate minimum to maximum, individual data points are overlayed. Black boxpots present control conditions RM and G-Krebs, green presents HC-CON and HC-BUT, blue presents IBS-CON and IBS-BUT, purple presents HC-FERCON and HC-FERFIB. Commonly used thresholds for biological relevance indicated by the gray dashed lines for 0.5- and 2-fold changes compared to G-Krebs. One-way ANOVA with posthoc uncorrected Fisher's LSD multiple comparisons test versus G-Krebs control condition. Exact *p*-value for ANOVA and if significant for posthoc tests provided in figure.

gene expression, tryptophan transport appeared unaffected. However, HC-FERFIB for the 24 h preventive measure increased tryptophan uptake, creating a pattern in disagreement with reduced gene expression of LAT1 and LAT2.

In contrast, as a treatment measure for stressed cells, these alterations were either absent or minimal. Serosal fluids evoked a few changes in gene and protein expression of only the minor tryptophan transporter LAT2, e.g. for HC-FERFIB ex vivo manipulations, but without altering tryptophan transport.

Generally, we observed few differences to be elicited by in vivo, intestinal manipulation (90-min butyrate infusion) serosal fluids (HC-BUT or IBS-BUT), with the IBS-derived samples being especially interesting since mucosal and fecal supernatants of such patients have been shown to cause nerve activations in the enteric nervous system[53–55]. Instead, particularly serosal fluids obtained from the 90-min ex vivo exposure to fecal fiber or control fermentation supernatants (HC-FERFIB or HC-FERCON) seemed to produce prominent, biologically relevant effects on all levels of evidence (gene/protein expression of LAT1 and LAT2, and tryptophan uptake).

Furthermore, based on the metabolite profiles of the different serosal fluids, we observed that the total SCFA concentration is comparable to

physiological levels in peripheral blood (80–180 μM)[56], yet at least 5-fold lower (~15 μM). However, the ratios are in agreement with the literature with acetate being the predominant SCFA. Although butyrate and propionate are usually untraceable in venous peripheral blood, they are found at concentrations of 1–15 μM in the systemic circulation, which are comparable to the measurements in serosal fluids of this study[57,58]. In particular, the serosal fluid HC-FERFIB had a visually higher content of SCFAs, while also showing a pattern of decreased tricarboxylic acid cycle metabolites. These observations may indicate that the increased control over experimental conditions during the ex vivo manipulations of the intestinal biopsies, as well as the consecutive exposure of the biopsies to these manipulations could lead to more consistent effects throughout the different levels of evidence (as opposed to more varied effects detected after in vivo manipulations), which can be captured using the current gut-brain axis model. Noteworthy is also, that not only SCFAs but also structural motifs of fiber fractions have recently been detected in fecal fermentation supernatants, prepared in an identical manner as the ones utilized for the ex vivo manipulations, and interacted differentially with human toll-like receptors[59].

As for the proinflammatory cytokine profile, all ten cytokines were at detection levels in all serosal fluids (see Supplementary Table 1). Cytokines

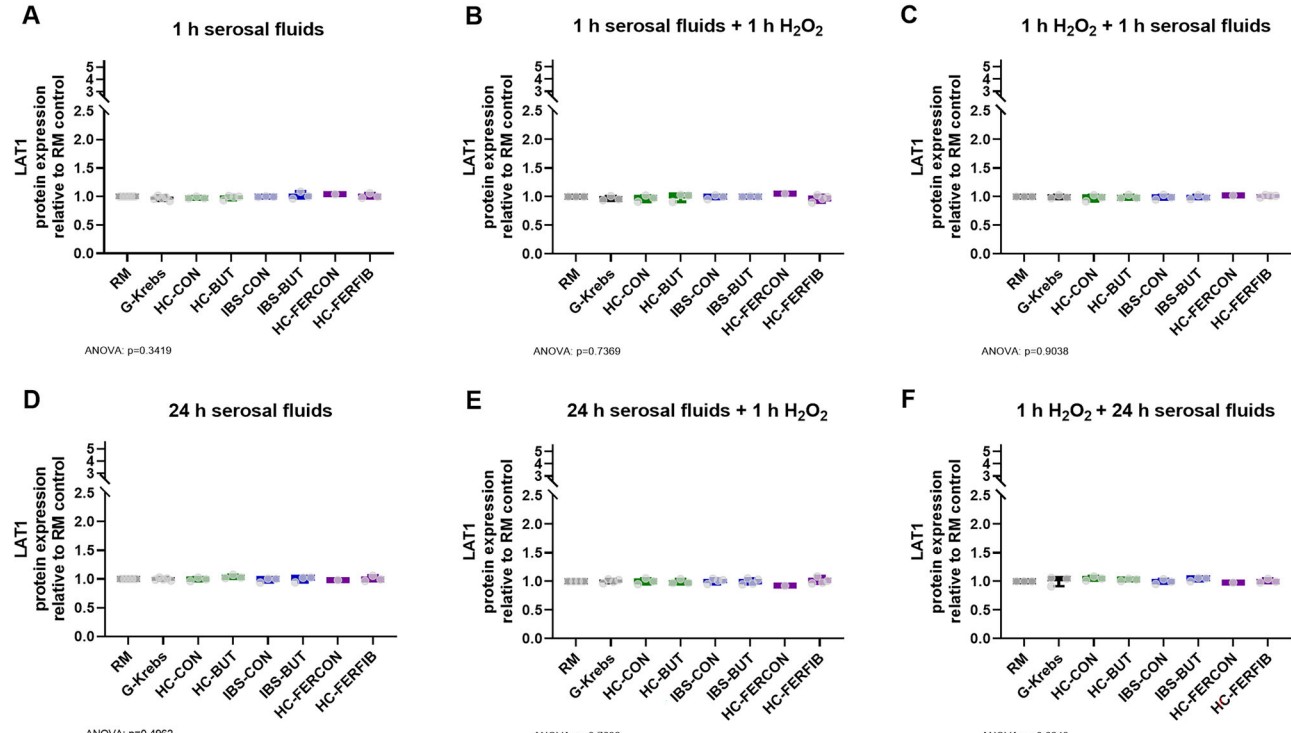

**Fig. 4 | Protein expression of the major tryptophan transporter LAT1 normalized to the methodological reference condition regular medium. A** 1 h exposure to serosal fluids alone, (**B**) exposure to 1 h serosal fluids as preventive measure before oxidatively stressing the cells (by 10 µM $H_2O_2$), (**C**) exposure to 1 h serosal fluids as a treatment measure after oxidatively stressing the cells, (**D**) 24 h exposure to serosal fluids alone, (**E**) exposure to 24 h serosal fluids as preventive measure before oxidatively stressing the cells, and (**F**) exposure to 24 h serosal fluids as a treatment measure after oxidatively stressing the cells. X-axis presents exposure to regular medium (RM), serosal fluid carrier solution G-Krebs, and serosal fluid pool derived from healthy biopsies collected from the unexposed colon (HC-CON), from healthy biopsies collected after in vivo butyrate exposure (HC-BUT), from irritable bowel syndrome biopsies collected from the unexposed colon (IBS-CON), from IBS biopsies collected after in vivo butyrate exposure (IBS-BUT), from healthy biopsies ex vivo exposed to supernatant of fecal control fermentation (HC-FERCON), and from healthy biopsies ex vivo exposed to supernatant of fecal fiber fermentation (HC-FERFIB). Per condition $n \leq 5$ biologically independent experiments. Data visualized as boxplots, horizontal line indicates median, whiskers indicate minimum to maximum, individual data points are overlaid. Black boxpots present control conditions RM and G-Krebs, green presents HC-CON and HC-BUT, blue presents IBS-CON and IBS-BUT, purple presents HC-FERCON and HC-FERFIB. One-way ANOVA with posthoc uncorrected Fisher's LSD multiple comparisons test versus G-Krebs control condition. Exact $p$-value for ANOVA and if significant for posthoc tests provided in figure.

were detected in all samples; even in the serosal fluid carrier solution (G-Krebs), and in the serosal fluids of the healthy participants. Interestingly, IBS-derived serosal fluids did not seem to have higher levels of proinflammatory cytokines, at least when visually compared to the serosal fluids of healthy participants (Supplementary Fig. 1). The concentrations measured in our experiment are comparable to values detected in serum levels of both young and older adults, encouraging further exploration of their potential contribution to the results we observed in this study on gene and protein expression of LAT1 and LAT2, as well as their activity, in fibroblasts[60].

Several studies on human dermal fibroblasts have demonstrated the capacity of different compounds with antioxidant properties (e.g., carotenoids, dietary polyphenols, estradiol, curcumin) to protect against subsequent exposure to $H_2O_2$, while failing to produce a significant rescue effect as post-stress treatments[61–63]. Contents in the serosal fluids with potential redox modulatory effects (*i.e.*, SCFAs) could act in a similar way[58,64], although it should be noted that we previously showed that directly adding sodium butyrate to the fibroblasts' culture media successfully restored oxidative stress-induced deficits of tryptophan transport by e.g., increasing the gene expression of the two tryptophan transporters LAT1 and LAT2[45].

In our previous study, high concentrations (mM) of butyrate alone had effects, while low concentrations (µM) did not. Both had treatment effects, though preventive effects were not tested[45]. Propionate (1 µM incubation for 12 h) showed preventive effects on efflux transporter activity and various expression patterns on human cerebro-microvascular endothelial cells

before lipopolysaccharide (LPS) exposure, *i.e.*, microbial stress, and protected from oxidative stress[65]. Also, tight junction expression and cytoskeletal arrangements were positively affected in murine brain endothelial cells exposed to butyrate or propionate with and without microbial stress (1 µM for 24 h, co-incubation with LPS for the second half)[66]. The effect of SCFAs has also been tested in a sequential system of four micro-physiological mesofluidic systems (the human 3xGLB system) including colonic epithelial cells from organoids (apical exposure to 20 mM in the ratio 6 acetate: 2 propionate: 2 butyrate for 4 days), peripheral blood mononuclear cells, a hepatocyte and Kupffer cell co-culture, and induced pluripotent stem cell (IPSC)-derived cerebral cells, the latter from a healthy and a Parkinson's disease donor—showing effects on gene expression patterns[37].

There have been few additional attempts to model the microbiota-gut-brain axis in vitro, all of which are unique in their setup and usually focus on the humoral signaling pathway. The MINERVA project adds fecal samples from a healthy individual and a patient with Alzheimer's disease to the microbiota compartment of a platform with five separate organ-on-a-chip devices including generic Caco-2 cells, macrophages and lymphocytes, endothelial cells and astrocytes, neurons and microglia cells[36]. Another attempt is neuroHuMiX which co-cultures *Limosilactobacillus reuteri* F275, Caco-2 cells and IPSC-derived enteric neurons in proximity allowing diffusion of secreted factors and metabolites via semipermeable membranes without direct cell-cell contact[67]. A dual-flow tissue perfusion device cultured an array of gastrointestinal and cerebral cell lines together with bacterial extracellular vesicles[68].

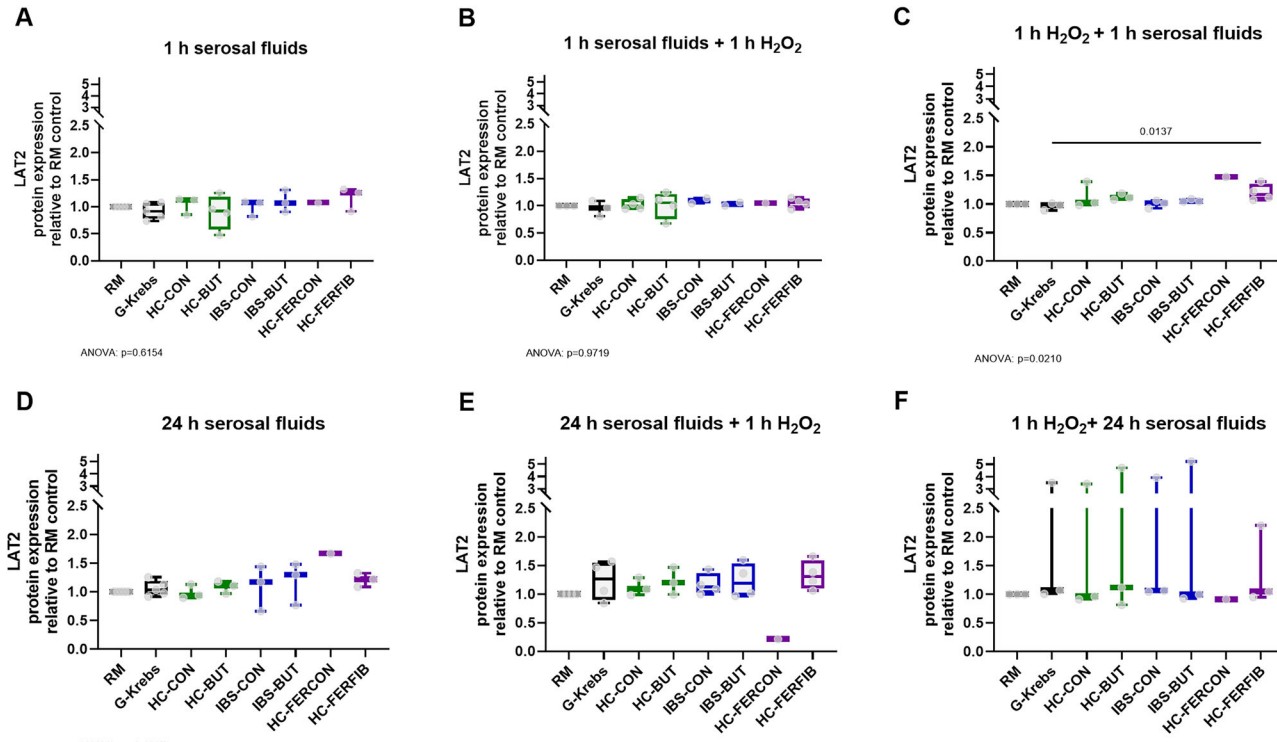

**Fig. 5 | Protein expression of the minor tryptophan transport LAT2 normalized to the methodological reference condition regular medium.** **A** 1 h exposure to serosal fluids alone, (**B**) exposure to 1 h serosal fluids as preventive measure before oxidatively stressing the cells (by 10 µM $H_2O_2$), (**C**) exposure to 1 h serosal fluids as a treatment measure after oxidatively stressing the cells, (**D**) 24 h exposure to serosal fluids alone, (**E**) exposure to 24 h serosal fluids as preventive measure before oxidatively stressing the cells, and (**F**) exposure to 24 h serosal fluids as a treatment measure after oxidatively stressing the cells. X-axis presents exposure to regular medium (RM), serosal fluid carrier solution G-Krebs, and serosal fluid pool derived from healthy biopsies collected from the unexposed colon (HC-CON), from healthy biopsies collected after in vivo butyrate exposure (HC-BUT), from irritable bowel syndrome biopsies collected from the unexposed colon (IBS-CON), from IBS biopsies collected after in vivo butyrate exposure (IBS-BUT), from healthy biopsies ex vivo exposed to supernatant of fecal control fermentation (HC-FERCON), and from healthy biopsies ex vivo exposed to supernatant of fecal fiber fermentation (HC-FERFIB). Per condition $n \leq 5$ biologically independent experiments. Data visualized as boxplots, horizontal line indicates median, whiskers indicate minimum to maximum, individual data points are overlayed. Black boxplots present control conditions RM and G-Krebs, green presents HC-CON and HC-BUT, blue presents IBS-CON and IBS-BUT, purple presents HC-FERCON and HC-FERFIB. One-way ANOVA with posthoc uncorrected Fisher's LSD multiple comparisons test versus G-Krebs control condition. Exact p-value for ANOVA and if significant for posthoc tests provided in figure.

As in all models, translating results from the lab bench to human physiology is a serious challenge. In our case, membrane transporters of dermal fibroblasts are comparable to BBB endothelial cells; however, we do not account for the actual microenvironment surrounding BBB structural cells. Systemic tryptophan, either free or albumin-bound, can be transported through specialized brain microvascular endothelial cells (BMECs) of the BBB by two closely related plasma membrane transporters, large neutral amino acid transporter 1 (LAT1) and 2 (LAT2)[69,70]. Commonly abundant in biological barriers, these transporters are also expressed in fibroblasts, making them a well-established in vitro model to study the effects of gut-derived substances on BBB physiology[46,47,71–74]. Importantly, fibroblasts were cultured to visual determined confluency of 95%, not creating a tight barrier. The herein presented work focused on the uptake of tryptophan. Future work should consider the cross-barrier transport, and could utilize transwell plate systems. For this reason, cell viability was assessed by morphological inspection (adhesive cells are alive, while floating cells or debris indicate cell death), and not measurements of transepithelial electrical resistance (TEER) prior to any experiment; and two molecular assays (AlamarBlue viability test and LDH cytotoxicity test) were performed for all conditions.

Similarly, colonic biopsies and Ussing chamber experiments are well-known approaches to study gut barrier function. Nevertheless, there are certain limits to maintaining the integrity of the biopsies and preserving the mucus layer on the luminal side of the biopsy, which is generally lost during processing, consequently eliminating its components from the current model. To ensure tissue viability, only biopsies with a potential difference <0.5 after reaching equilibrium in the Ussing chamber were used for experiments and subsequent collection of serosal fluids. The lowest, in any individual Ussing chamber, measured TEER was $9\,\Omega\,cm^{-2}$. For further details, please refer to Scharf et al. 2025[75]. As discussed above, the Ussing chamber creates an open, non-sterile environment. Several of the herein presented results indicate that samples collected from this system nevertheless seem to be suitable to be added to cell cultures.

Serosal fluids have been collected from recently conducted human trials that focused on other primary objectives. Available volumes of serosal fluids were limited, which allowed only a small number of biologically independent experiments, and limited statistical conclusions. While the colonic in vivo exposure to butyrate was conducted in both HC and IBS, the ex vivo exposure to supernatants from fecal fiber fermentations was solely conducted in HC. Although the assessment of serosal fluids from IBS biopsies ex vivo exposed to supernatants from fecal fiber fermentations would have been a strong addition to this work, such samples were not available, but should be considered in the future. As previously reported by Scharf et al. 2025[75], IBS symptoms were assessed using the Gastrointestinal Symptom Rating Scale (GSRS) for IBS for the week before sample collection, and averaged at 3.8 ± 0.8 which indicates moderate discomfort (on a scale from 1−7 for no to very severe discomfort) and is typical for IBS cohorts. Upon asking on the day of sample collection, none of the participants reported an alteration in their medical condition. However, this was a short-term observation and not compared to long-term IBS symptomatology.

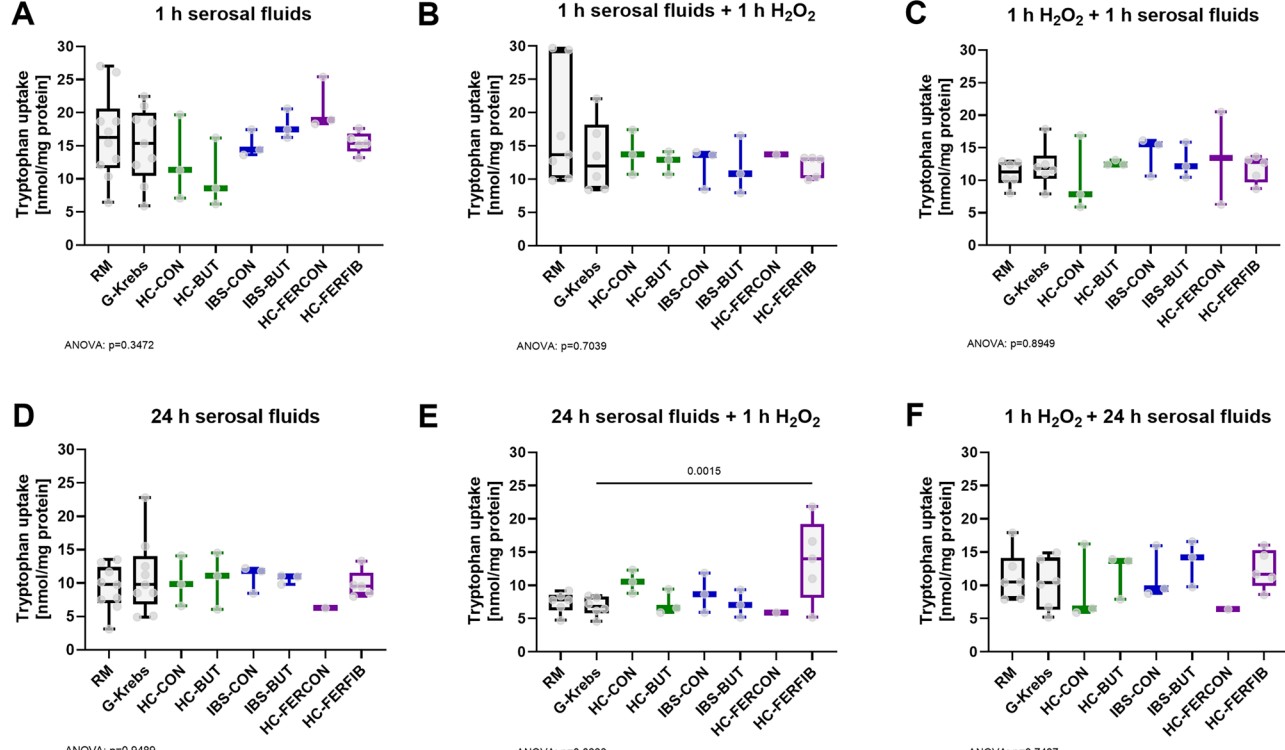

**Fig. 6 | Tryptophan uptake as an assessment of protein activity of transmembrane tryptophan transporters. A** 1 h exposure to serosal fluids alone, (**B**) exposure to 1 h serosal fluids as preventive measure before oxidatively stressing the cells (by 10 μM $H_2O_2$), (**C**) exposure to 1 h serosal fluids as a treatment measure after oxidatively stressing the cells, (**D**) 24 h exposure to serosal fluids alone, (**E**) exposure to 24 h serosal fluids as preventive measure before oxidatively stressing the cells, and (**F**) exposure to 24 h serosal fluids as a treatment measure after oxidatively stressing the cells. X-axis presents exposure to regular medium (RM), serosal fluid carrier solution G-Krebs, and serosal fluid pool derived from healthy biopsies collected from the unexposed colon (HC-CON), from healthy biopsies collected after in vivo butyrate exposure (HC-BUT), from irritable bowel syndrome biopsies collected from the unexposed colon (IBS-CON), from IBS biopsies collected after in vivo butyrate exposure (IBS-BUT), from healthy biopsies ex vivo exposed to supernatant of fecal control fermentation (HC-FERCON), and from healthy biopsies ex vivo exposed to supernatant of fecal fiber fermentation (HC-FERFIB). Per condition $n ≤ 10$ biologically independent experiments. Data visualized as boxplots, horizontal line indicates median, whiskers indicate minimum to maximum, individual data points are overlayed. Black boxpots present control conditions RM and G-Krebs, green presents HC-CON and HC-BUT, blue presents IBS-CON and IBS-BUT, purple presents HC-FERCON and HC-FERFIB. One-way ANOVA with posthoc uncorrected Fisher's LSD multiple comparisons test versus G-Krebs control condition. Exact p-value for ANOVA and if significant for posthoc tests provided in figure.

While the effect of long-term diet on the results cannot be ruled out, the last meal before a fasting period prior to colonoscopy was standardized to low fiber intake for all participants.

Moreover, interactions between the involved systems (microbiota, gut and blood-brain barrier) are sequential in the presented model, more similar to functional coupling as defined and tested by Vernetti et al., which uses five topologically separate micro-physiological systems including primary intestinal cells and a BBB/neuron compartment[76]. In comparison, other gut-brain axis models have a fluidic organ-on-a-chip structure, which offers different temporal and topological information[34,36,38,67,68].

Our approach is based on well-established methodologies, both on the intestinal and the brain side, focusing on the humoral route of gut-brain communication, and providing plenty of opportunities to be extended as well as to be applied bidirectionally. This highlights the humanized nature of this model, assembled by elements isolated directly from the human body and that are specific to the individual providing them, including the possibility to be linked to individual disease background. Namely, fibroblasts can easily be isolated from individuals for more personalized investigations in parallel with their colonic biopsies and serosal fluids. The well-documented use of fibroblasts allows for experimentation with more complex cultures, either three-dimensional or co-cultures[77,78]. Also, dermal fibroblasts can be reprogrammed into IPSCs[34,37,40,79], giving our model the possibility to evolve and increase similarity with BBB characteristics or to expand by adding an ex vivo approach of the neuronal system. Taken together, our model opens possibilities for assessments on an individual level, hence supporting precision nutrition or medicine.

To our knowledge, there have only been a few first attempts to build a generalized gut-brain axis model ex vivo, allowing more detailed mechanistic assessments than sole in vivo examinations[33]. Our humanized model can efficiently be used to detect effects on all levels—gene expression, protein expression, protein activity, and metabolite profiles. While serosal fluids alone did not exert effects, serosal fluids as a preventive measure (which is the physiologically most relevant one) exerted more effects than as a treatment measure. Generally, serosal fluids originating from ex vivo modification of the intestinal environment had more prominent effects on tryptophan uptake, likely due to methodological differences, than those of in vivo manipulation. Consequently, ex vivo approaches may be more relevant to study with the current gut-brain axis model.

## Methods
### Collection of intestinal serosal fluids
Serosal fluids were sampled from Ussing chamber systems containing human intestinal tissue (similar procedure described by Tabat et al. 2020[26]) that have been A) in vivo exposed to butyrate—hereafter named BUT, or B) ex vivo, mucosally exposed to fecal-fermented fiber fractions extracted from a butyrate-promoting wheat bran (arabinoxylans and xylo-oligosaccharides, with or without ferulic acid)—hereafter named FERFIB, or respective controls; both of these conditions mimic a fiber-rich diet. Serosal fluids for fibroblast exposure were available from recently conducted human trials

with healthy adults (HC, part A and B) and adults with irritable bowel syndrome (IBS, part A only), ages 18–65 years.

Part A) For the in vivo butyrate exposure in HC and IBS[75], a double-balloon colonoscopy setup, without prior cleansing of the bowel, was used to expose a well-defined area of the distal colon to 60 mL of 100 mmol/L sodium butyrate solution for 90 min. The two gently inflated latex balloons were placed ~20–25 cm apart from each other in the descending colon. The administered butyrate solution was held in between the two balloons in the targeted area throughout the time of exposure. Mucosal biopsies were collected prior to the butyrate exposure just proximally of the targeted colonic area (CON) and after the exposure within the targeted and exposed area (BUT). The biopsies were subsequently mounted in Ussing chambers (maintained at 37 °C and oxygenized) for 60 min. The Ussing chamber is a well-established method to assess intestinal transport and permeability[80]. In this system, a freshly harvested intestinal biopsy is mounted between two chamber halves separating the luminal and serosal sides, under close to physiological conditions. Both chamber halves are filled with Krebs solution at experiment start.

Part B) Healthy adults (HC) donated fresh fecal samples for fecal fermentation of four different fiber fractions and underwent distal colonoscopy to collect colonic biopsies. These biopsies were, in the Ussing chambers, ex vivo exposed for 90 min to the supernatant of ex vivo fecal fiber fermentation (FERFIB) or supernatant of fecal control fermentation (FERCON). Of note, each subject's biopsies were exposed to their own fecal material fermented with different wheat bran fiber fractions. To obtain the fecal fermentation supernatants, wheat bran fiber fractions were homogenized and subjected to simulated in vitro digestion and dialysis. The fermentation medium used was as outlined by Fooks and Gibson[81] replacing glucose with the freeze-dried fiber fractions (5% w/v) as a carbon source. After inoculation with 20% fecal slurry, fermentations were conducted at 37 °C under anaerobic conditions for 24 h. Fermentation supernatants were collected and centrifuged for 10 min at 15,000 x g, 4 °C and frozen at -80 °C. Supernatants were defrosted and added to the mucosal Ussing chambers at ~6–7% v/v.

All colonoscopies were performed in the morning after an overnight fasting period. More than 10 h prior to colonoscopy, all participants were provided with a standardized low-fiber evening meal (pasta and tomato sauce).

The major hurdle of this study was the limited volume of serosal fluids. Approximately 600–700 μL of serosal fluid could be collected per Ussing chamber. To facilitate the number of different experiments, serosal fluids of all chambers of all subjects of a given type were pooled in equal amounts, resulting in the following serosal fluid pools: derived from IBS biopsies collected from the unexposed colon (IBS-CON), from IBS biopsies collected after in vivo butyrate exposure (IBS-BUT), from healthy biopsies collected from the unexposed colon (HC-CON), from healthy biopsies collected after in vivo butyrate exposure (HC-BUT), from healthy biopsies ex vivo exposed to supernatant of fecal fiber fermentation (HC-FERFIB), and from healthy biopsies ex vivo exposed to supernatant of fecal control fermentation (HC-FERCON).

### Characterization of serosal fluids

Assessment of microbial growth of aerobic bacteria was based on the cultivation on trypticase soy agar and blood agar for 3 days at 37 °C. Protein concentration was assessed by Bradford assay, and quantification of ten proinflammatory cytokines was performed using an MSD® multi-spot assay system. Additionally, the pH was assessed by pH meter and further validated using pH indicator paper. The metabolite profiling of serosal samples was conducted using liquid chromatography coupled to time-of-flight high-resolution mass spectrometry (UHPLC-qToF-HRMS).

### Quantification of proinflammatory cytokines in serosal fluids.

To measure proinflammatory cytokines in the different serosal fluids the Meso Scale Discovery V-Plex® Proinflammatory Panel 1 (human) kit was used (LOT: Z0047920, Meso Scale Diagnostics LLC). The 96-well, 10-spot plate was prepared according to the manufacturer's instructions and no further dilutions were required for the serosal fluids, which were used in a volume of 50 μL per well in technical triplicates. The MSD® plate was washed with 200 μL of the assay's wash buffer (supplied by the manufacturer) according to the instructions. During incubations the plate was placed on a plate shaker at 600 rpm.

### Analysis of metabolite profiles of serosal fluids.

An aliquot (25 μL) of the extract obtained for metabolite profiling was derivatized using 3-nitrophenylhydrazin (3-NPH) for the ultra-high performance liquid chromatography coupled to time-of-flight high-resolution mass spectrometry (UHPLC-qToF-HRMS) analysis of short-chain fatty acids (SCFAs) and tryptophan metabolites, similar as described by Seeburger et al. 2023[82]. Briefly, the sample was mixed with 25 μL of 50 mM 3-NPH, 25 μL of 50 mM N-ethyl carbodiimide and 25 μL of 7% pyridine in methanol (MeOH):water (7:3, v/v) spiked with 1 μg/mL of the internal standards (ISs) acetate-d4, butyrate-d8 and propionate-d8. Four extraction blanks were also prepared following the same procedure above but adding 25 μL of 0.9% sodium chloride (NaCl) instead of sample. The mixture was incubated on an orbital shaker (300 rpm) for 1 h at room temperature and filtered through a 0.22 μm membrane. Four pooled samples were also prepared as quality controls (QC) by mixing equal aliquots of each sample for further performance analysis.

UHPLC-qToF-HRMS analysis was performed using a 1290 Infinity UHPLC (Agilent) interfaced with a dual electrospray ionization (ESI) source to a 6545 qToF-MS system (Agilent) and equipped with an Acquity BEH C18 column (2.1 × 100 mm, 1.7 μm; Waters Corporation). Mobile phase (MP)-A 0.1% formic acid in water and MPB acetonitrile were eluted at 0.4 mL/min starting with 10% MPB (0–2 min), 0–100% MPB (2–4 min), 100% MPB (4–6 min), followed by re-equilibration with 10% MPB for 4 min. The column temperature was maintained at 50 °C, while the autosampler was maintained at 10 °C. Accurate mass spectra were acquired (2 spectra/s) with an $m/z$ range of 70–1500 in negative ion mode. The dual ESI source was programmed as follows: collision energy 0 V, capillary voltage 3.6 kV, nozzle voltage 1500 V, and $N_2$ pressure at nebulizer, flow rate and temperature as sheath gas set at 21 psi, 10 L/min and 379 °C, respectively. The injection volume was 5 μL. MassHunter Workstation Software (Agilent) was used for data acquisition and processing.

Data pre-processing was performed using the ADAP pipeline within MZmine 2.53 software as described by Seeburger et al. 2023[83]. The pre-processing steps included mass detection, extraction of ion chromatograms, peak deconvolution/integration, isotopic peak grouping, sample alignment, and gap filling. Identification of features was performed by comparing retention time (RT) and accurate mass-to-charge ratio (5 ppm) of features to that of analytical standards. The peak areas of the metabolites were normalized to the IS closest in RT. The relative standard deviation (%RSD) of IS in both samples and QC was satisfactory (< 30%) (Supplementary Table 2), meeting standard requirements for metabolomics analysis[84]. Concentration of SCFAs was calculated against six-points external calibration curves (0.1–50.0 μg/mL) of acetate ($y = 668.54x \pm 108.69$, $r^2 = 0.9945$), propionate ($y = 3580.13x \pm 127.59$, $r^2 = 0.9966$), butyrate ($y = 3100.18x \pm 403.6$, $r^2 = 0.9942$), valerate ($y = 207.67x \pm 0.01$, $r^2 = 0.9990$) and caproate ($y = 147.66x \pm 88.94$, $r^2 = 0.9938$), while data for other metabolites including tricarboxylic acid cycle (TCA)-related metabolites (lactate, ketoglutarate, pyruvate, and succinate) as well as the tryptophan (TRP)-related metabolites (indole-3-acetic acid (IAA), indole-3-propionic acid (I3PA), indole-3-butyric acid (I3BA), indole-3-carbinol (I3C), and kynurenic acid) were expressed as normalized peak area.

### Exposure of fibroblasts to intestinal serosal fluids

Commercially available human dermal fibroblast cells (ATCC-PCS-201-012, American Type Culture Collection) were cultured under adequate conditions, as described previously[45]. In brief, cells (up to passage 12) were cultured in minimum essential medium containing 10% fetal bovine serum, 1% penicillin, 1% streptomycin, and 1% L-glutamine (all Gibco Cell Culture,

Thermo Fisher Scientific) at 37°C and 5% $CO_2$. When cells reached 95% confluency (visually determined), they were in vitro exposed to the following (also described in Table 1):

– Control conditions
– Oxidative stress, evoked by hydrogen peroxide ($H_2O_2$) incubation
– Serosal fluids of human colonic biopsies:
   • mucosally exposed to fecal fermentation supernatant without fibers ex vivo (FERCON); healthy
   • mucosally exposed to fecal fermentation supernatant with fibers ex vivo (FERFIB); healthy
   • before butyrate exposure in vivo (CON); IBS and healthy
   • after butyrate exposure in vivo (BUT); IBS and healthy
– Oxidative stress + serosal fluids—to assess treatment potential
– Serosal fluids + oxidative stress—to assess preventive potential

Since the serosal fluids' carrier solution was physiological glucose-Krebs (G-Krebs), it was low on several nutrients required for adequate fibroblast culture. Therefore, we finally exposed fibroblasts to ½ serosal fluid and ½ regular cell culture medium of total volume per well (Table 1).

Oxidative stress was induced by 1 h incubation with 10 μM hydrogen peroxide ($H_2O_2$ 30%; Sigma-Aldrich) in regular medium.

### Outcome measurements

**Cell viability and cytotoxicity assays.** Cell viability of fibroblasts after different exposures was determined by AlamarBlue, CyQUANT™ LDH Cytotoxicity assay and microscopical examination of morphology.

Fibroblasts cultured in 96-well plates according to the experimental conditions presented in Table 1 were used for cell viability assays (AlamarBlue™ Cell Viability Reagent, Invitrogen, Thermo Fisher Scientific), and for lactate dehydrogenase (LDH) cytotoxicity assays (CyQUANT LDH Cytotoxicity Assay™, Invitrogen, Thermo Fisher Scientific). Experiments were conducted in technical triplicates and with a total volume of 150 μL each and incubated at 37°C (5% $CO_2$). For the last hour of incubation, also control wells were prepared: For cell viability, addition of 10 μL dH$_2$O to 140 μL RM per well was used as a positive control treatment, whereas addition of 10 μL Lysis Buffer acted as the negative control. For the LDH cytotoxicity assessment, the meanings of positive and negative controls were inverted.

After incubation, 60 μL of the supernatant was pipetted into a new 96-well plate to perform the cytotoxicity assay according to the manufacturer's instructions. Absorbance at 680 nm was subtracted from the one measured at 490 nm.

For the cell viability assay, 10 μL of AlamarBlue™ was added to the remaining 90 μL medium in each well of the original 96-well plate, which was then incubated for 55 min at 37°C (5% $CO_2$). Absorbance at 600 nm was subtracted from the absorbance measured at 570 nm for each well.

**Metabolite profiling of cell supernatants and harvest.** Since the AlamarBlue™ used in the above assay is considered non-toxic to the cells and there is no lysis step, supernatants were removed from the wells and stored at -80°C for metabolomics analyses, while cells were trypsinized

with 20 μL trypsin-EDTA per well (0.25%, with phenol red; Gibco Cell Culture, Thermo Fisher Scientific) to be harvested. After adding the trypsin, the plate was shaken at 300 rpm for 1 min and the cell suspension from each well was transferred to an Eppendorf tube for centrifugation at 1250 x g for 5 min. Cells were resuspended in 200 μL Dulbecco's Phosphate-Buffered Saline (DPBS; Gibco Cell Culture, Thermo Fisher Scientific) and were centrifuged once more at 1250 x g for 5 min, before being resuspended in 50 μL DPBS. The three wells of cell supernatants and cell harvest, respectively, per condition of all experiments were pooled and stored at -80°C until further analysis.

Extraction of cell harvest (cell lysate) and cell supernatant (conditioned medium) was performed in deep well plates by mixing 20 μL of sample with 240 μL of a monophasic solvent system composed of methanol, methyl-tert butyl ether, and isopropanol (MeOH:MTBE:IPA, 4:3:3) spiked with 1 μg/mL of the following ISs: hexanoic acid-d3, succinate-d4, tryptophan-d5, glycoursodeoxycholic acid-d4 (GUDCA-d4), cholic acid-d4 (CA-d4), litocholic acid-d4 (LCA-d4), ursodeoxycholic acid-d4 (UDCA-d4), glycocholic acid-d4 (GCA-d4), chenodeoxycholic acid-d4 (CDCA-d4), glycodehydrocholic acid-d4 (GDCA-d4), taurocholic acid-d4 (TCA-d4), glycolitocholic acid-d4 (GLCA-d4), deoxycholic acid (DCA d4), and heptadecanoic acid. Four extraction blanks were also prepared by mixing 40 μL of 0.9% NaCl with the monophasic solvent system. The mixture was incubated on an orbital shaker (300 rpm) for 1 h at 4°C and filtered through a 0.22 μm membrane. An aliquot of the resulting extract (100 μL) was concentrated under vacuum for 1 h at ambient temperature (Concentrator Plus, Eppendorf) and reconstituted in 50 μL MeOH:water (7:3, v/v) for analysis by UHPLC-qToF-HRMS. Four pooled samples were also prepared as QCs by mixing equal aliquots of each sample for further performance analysis.

UHPLC-qToF-HRMS analysis was performed using an Xevo G3 qToF system (Waters Corporation) equipped with a Acquity BEH C18 column (2.1 × 100 mm, 1.7 μm, Waters Corporation). Accurate mass spectra were acquired (5 spectra/s) with a m/z range of 50–1200 in negative ionization mode. The ESI source was set with the following settings: capillary voltage set at 1.5 kV, collision energy at 0 V, sampling cone at 40 V, source temperature at 150°C, cone gas flow at 1 L/min and desolvation gas flow and temperature set at 16 L/min and 550°C, respectively. The MPA consisted of 2 mM ammonium acetate (NH$_4$Ac) in MeOH:water (7:3, v/v) and MPB of 2 mM NH$_4$Ac in MeOH. The MP eluted with 0.4 mL/min by the following gradient: 10% MPB (0–2 min), 0–100% MPB (2–4 min), 100% MPB (4–6 min) followed by a re-equilibration with 10% MPB (4 min). The injection volume was 1 μL and the temperature of the column and autosampler was 50°C and 10°C, respectively.

Data pre-processing was performed using the ADAP pipeline within MZmine 2.53 software as described above for the serosal fluid characterization (SCFA analysis). Identification of features was performed by comparing RT and accurate mass-to-charge ratio (3 ppm) of features to that of analytical standards of amino acids (AA), bile acids (BA), fatty acids (FA), tricarboxylic acid cycle-related metabolites (TCA), tryptophan-related metabolites (TRP) and other metabolites (OTH, including sugars and organic acids) from an in-house library. Peak corresponding to the same

### Table 1 | Conditions for exposure

| Control condition | RM | 1 or 24 h regular cell culture medium |
|---|---|---|
| Control condition | Glucose-Krebs (G-Krebs) | 1 or 24 h ½ RM and ½ G-Krebs solution (serosal fluid's carrier solution) |
| Oxidative stress | 10 μM $H_2O_2$ | 1 h RM with 10 μM $H_2O_2$ |
| Serosal fluids alone | Serosal fluid | 1 or 24 h ½ RM and ½ serosal fluid (*i.e.*, HC-CON & HC-BUT, IBS-CON & IBS-BUT, HC-FERCON & HC-FERFIB) |
| Prevention | Serosal fluid + 10 μM $H_2O_2$ | 1 or 24 h serosal fluid then 1 h RM with 10 μM $H_2O_2$ |
| Treatment | 10 μM $H_2O_2$ + serosal fluid | 1 h RM with 10 μM $H_2O_2$ then 1 or 24 h serosal fluid |

Serosal fluid pools derived from healthy biopsies collected from the unexposed colon (HC-CON), from healthy biopsies collected after in vivo butyrate exposure (HC-BUT), from IBS biopsies collected from the unexposed colon (IBS-CON), from IBS biopsies collected after in vivo butyrate exposure (IBS-BUT), from healthy biopsies ex vivo exposed to supernatant of fecal control fermentation (HC-FERCON), and from healthy biopsies ex vivo exposed to supernatant of fecal fiber fermentation (HC-FERFIB).

elution pattern and m/z of xylose and glucose were defined as a non-identified pentose (NI pentose) and hexose (NI Hexose), respectively, as the applied method is not able to distinguish isomeric monosaccharides. After pre-processing, features with relatively high abundances in blanks were removed, and the peak areas of the remaining features were normalized to the IS closest in RT. QC samples were evenly distributed throughout the UHPLC-qToF-MS analysis batch and only features with a %RSD < 30% in QC samples were retained in the final dataset. Similarly to SCFA analysis, the %RSD of IS in both samples and QC was satisfactory (< 30%) (Supplementary Table 3), meeting standard requirements for metabolomics analysis[84].

Supplementary Table 4 shows the metabolites included in the analyses of serosal fluid, G-Krebs, cell harvest, and supernatants, along with their subclass, molecular formula and monoisotopic mass (Da).

**Assessment of the tryptophan transport system**. The tryptophan transporter LAT1 is encoded by the *SLC7A5* gene and enables the bidirectional transport of branched or aromatic amino acids in a sodium- and gradient-dependent manner. Its translocation and function on the cell membrane depends on the membrane glycoprotein 4F2 heavy chain (4F2hc), which is derived from the expression of the *SLC3A2* gene. LAT2 is encoded by the *SLC7A8* gene and functions similarly as LAT1, binding with neutral amino acids[69,70,85].

Gene expression of *SLC7A5*, *SLC7A8*, *SLC3A2* and reference gene *GAPDH* (which encodes glyceraldehyde 3-phosphate dehydrogenase) was measured with quantitative polymerase chain reaction (qPCR). Protein expression of LAT1 and LAT2 was assessed by enzyme-linked immunosorbent assays. For protein activity/functionality, transmembrane transport of amino acids into the fibroblasts was measured using a radioactive tracer (*i.e.*, ³H-L-tryptophan).

**RNA and protein extraction**. Fibroblasts were cultured in 6-well plates with 2 mL total volume per well for RNA and protein extraction using the AllPrep RNA/Protein kit (Qiagen). The AllPrep kit allowed for the RNA and proteins to be isolated simultaneously during one assay. Following the experimental incubations presented in Table 1, plates were allowed to reach room temperature before removing the supernatant from each well. Each well was briefly washed with 1 mL DPBS and immediately after, each well was treated with 200 μL APL buffer (Qiagen). The plate was incubated for 5 min at room temperature to allow the cells to detach from the culturing surface. After incubation, a cell scraper was used to mechanically detach remaining cells from the walls and bottom of the well. The lysates were then pipetted into a 1.5 mL RNase-free tube and were subjected to aspiration for minimum 30 times using a 20 G needle (BD Eclipse™) and a 5 mL syringe, to homogenize the samples. Lysates were then placed on ice and were used to extract proteins and RNA according to the AllPrep kit protocol. The final volume of protein samples was approximately up to the originally added 200 μL of APL Buffer. RNA extracts were run twice through the spin column, diluted in 15 μL RNAse-free water included in the kit.

Nanodrop 2000C (Thermo Fisher Scientific) was used for RNA and protein quantification, and sample purity was assessed based on the 260/280 ratio. For protein, a ratio of 0.6-0.8 is generally considered pure, while for RNA, a ratio of ~2.0 is generally acceptable. Protein and RNA samples were kept on ice and then immediately stored at -80°C for future processing. RNA concentrations and RNA integrity number (RIN) were further assessed using Agilent 2100 Bioanalyzer (Agilent). RIN values range from 1−10, with values close to 1 indicating low RNA integrity due to degradation by RNases, whereas values closer to 10 indicating high RNA integrity. The level of RNA degradation is determined based on the 18S and 28S ratio of ribosomal RNA in eukaryotic cells.

**Gene expression**. RNA was inversely transcribed into cDNA by using PCR and SuperScript™ VILO™ cDNA synthesis kit (Invitrogen, Thermo Fisher Scientific) on the 2700 Thermal Cycler (Applied Biosystems by Life

Technologies, Thermo Fisher Scientific); 25°C for 10 min, 42°C for 120 min, 85°C for 5 min, 25°C for 5 min. The RNA concentrations determined by Bioanalyzer were considered for loading 100 ng RNA into cDNA synthesis. The gene expression differences of *SLC7A5*, *SLC3A2*, and *SLC7A8* were determined in technical triplicates by using a QuantStudio 7 Flex fast real-time PCR system (Applied Biosystems by Life Technologies, Thermo Fisher Scientific); 95°C for 20 s, 40 × 95°C for 1 s, 60°C for 20 s) with primers and hydrolysis probes of TaqMan gene expression assay (Hs00185826_m1 SLC7A5, Hs00374243_m1 SLC3A2, Hs00794796_m1 SLC7A8; Applied Biosystems, Thermo Fisher Scientific) and LuminoCt qPCR ReadyMix kit (Sigma-Aldrich). Glyceraldehyde 3-phosphate dehydrogenase (*GAPDH*, Hs03929097_g1 GAPDH; Applied Biosystems, Thermo Fisher Scientific) was used as a reference gene in this experiment because of the stable mRNA expression of *GAPDH* in human diploid fibroblasts under different treatment and cell conditions. Relative quantification of gene expression was performed by comparing the number of cDNA copies of treated cells and controls using the comparative Ct method ($\Delta\Delta$Ct): $2^{-\Delta\Delta Ct}$ was used to calculate the quantification of mRNA; $\Delta$Ct = Ct target − Ct GAPDH; and $\Delta\Delta$Ct = $\Delta$Ct test sample − $\Delta$CT regular medium. Hence, the fold-change gene expression was based on methodological reference, *i.e.*, regular medium-incubated cells.

For our samples, we considered an RNA quality with RIN > 5 and A260/280 1.8-2.2 as sufficient. Experimental batches with >50% of samples not meeting those quality standards, were not processed further. Therefore, gene expression of the 1 h preventive and treatment conditions was not assessed.

**Protein expression**. Protein expression of LAT1 and LAT2 were assessed in technical triplicates per sample using enzyme-linked immunosorbent assays (ELISAs) (MBS9938814 and MBS9365819, MyBioSource Inc.) according to the manufacturer's protocol. Protein expression was normalized per experiment as: protein expression in a test sample per protein expression in the methodological reference, *i.e.*, regular medium-incubated cells. Absorbance was measured using a Cytation 3 Imaging Reader (BioTek).

**Tryptophan transporter activity**. Fibroblasts were cultured in 24-well plates for tryptophan transporter assays. Experiments were conducted in technical duplicates and with a total volume of 500 μL each and incubated at 37°C (5% $CO_2$). After incubation with the experimental conditions, fibroblasts were washed with DPBS and amino acid transport assays were carried out by using 20 μL of ³H-L-tryptophan (30 Ci/mmol; Larodan Fine Chemicals AB) and 180 μL of 0.56 mM unlabeled tryptophan (Sigma-Aldrich). After 5 min of incubation at 37°C, the cells were rapidly washed twice with ice-cold DPBS and lysed by incubating for 30 min with 200 μL of 0.5 M sodium hydroxide (Sigma-Aldrich). The cell lysate was used to quantify the cell protein in each well by Bradford reagent (Sigma-Aldrich) using bovine serum albumin as a standard. The radioactivity of the cell lysate was measured by a liquid scintillation analyzer (TRI-CARB 2100TR, Packard Instrument Company) from a mixture of scintillation cocktail (Optiphase, Hisafe 3, Perkin Elmer Life Sciences) and cell lysate.

Tryptophan uptake was normalized to total protein. Experiments were conducted in two batches; hence results were normalized based on the mean tryptophan transport per protein in the methodological reference, *i.e.*, regular medium-incubated cells.

For a discussion on the technical aspects of the individual methods see Supplementary Note 1.

**Statistics and reproducibility**
Since this was a proof-of-concept study with limited *prior* data, an exact sample size calculation was not feasible. We used serosal fluids from $n > 10$ subjects per condition to be analyzed in replicates each. Gene and protein expression were typically assessed in $n \leq 5$, and transport assays typically in $n \leq 10$ biologically independent experiments (each using the

pooled serosal fluids of $n > 10$ subjects), and in technical replicates, as described per outcome measure above.

Statistical analyses were conducted in GraphPad Prism (version 10.2.2). Data for cell viability, cytotoxicity, gene and protein expression, as well as for tryptophan uptake were assessed for normality using Shapiro-Wilk tests. Robust regression and outlier removal (ROUT) at $Q = 10\%$ was used. One-way analysis of variance (ANOVA) was performed with *posthoc* uncorrected Fisher's least significant difference (LSD) multiple comparisons, two-sided. Whenever the ANOVA resulted in $p < 0.1$, multiple comparisons were plotted and significant differences were indicated with exact *p*-values if $p < 0.05$ in the graphs plotted with GraphPad Prism. In the text, effect sizes are reported as mean difference ± standard error of difference, together with exact *p*-value and F-value (degrees of freedom between groups, degrees of freedom within groups).

Due to the proof-of-concept nature of this study, and with planned independent comparisons limited against the G-Krebs control, descriptive *p*-values were reported, and focus was laid on effect sizes and patterns of differences on the various levels of evidence (gene and protein expression, protein activity). A fold-change lower than 0.5- or higher than 2-fold compared to G-Krebs was selected as a cutoff for biologically relevant results, as commonly used. Regular medium exposures were used as methodological reference but not included in the statistical comparisons.

Furthermore, metabolite profiling and quantification of pro-inflammatory cytokines of the serosal fluids (G-Krebs included), cell harvest and supernatants were conducted for exploratory purposes. Results were based on analyses of the pooled sample, in singlets for metabolite profiling and in triplicates for the cytokine profiling. Metabolite profiles of cells and supernatants were visualized as a heatmap built in R v.4.4.3 using package *pheatmap*[86,87] based on log2 fold changes between stress and serosal fluid alone exposures for 1 h and 24 h, respectively. Results for SCFA and cytokine concentration for serosal fluids and G-Krebs were plotted using GraphPad Prism. Lastly, results for SCFAs, tricarboxylic-acid cycle (TCA)-metabolites, and tryptophan (TRP) metabolites are presented as heatmaps using GraphPad, showing the log2 fold changes between G-Krebs and serosal fluids.

### Ethical considerations

Serosal fluids were obtained from two recently conducted human trials. The study for part 1 (HC-CON, HC-BUT, IBS-CON, IBS-BUT) was approved by the Central Ethical Review Board, Stockholm, Sweden and the Swedish Ethical Review Authorities (registration numbers 2016/464/1, 2023-01125-02). The study for part 2 (HC-FERCON, HC-FERFIB) was approved by the Swedish Ethical Review Authorities (registration numbers 2020-03943, 2020-06884 and 2023-03495-02). Studies were conducted in accordance with the Declaration of Helsinki and its amendments. Written informed consent was obtained from all study participants. All ethical regulations relevant to human research participants were followed.

### Reporting summary

Further information on research design is available in the Nature Portfolio Reporting Summary linked to this article.

### Data availability

Numeric source data for Figs. 1–6 are provided in the Supplementary Data 1 file.

### Code availability

No codes or custom-designed algorithms were used for the conduction of this work.

### Biological material availability

All unique biological materials (e.g. serosal fluids) were collected for the purpose of the study and have been used up, and are hence not available.

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

## Acknowledgements

This study was conducted within the Nutrition-Gut-Brain Interactions Research Centre (NGBI) and the Responsive Nutrition Profile (Rosetta@ORU) at Örebro University. We would like to thank our previous MSc student Lotte Smit and BSc student Alicia Blåder for their piloting work on this project. We would like to thank postdoctoral researcher Anh Hoang Nguyen and BSc student Lovisa Trege for their experimental metabolomics work. We would like to thank our previous research assistant Linnea Brengesjö Johnson for assisting in some lab work. The establishment of the model system based on the samples from healthy individuals was funded by a Research Fellowship (2020–2023) awarded to Julia Rode by the European Society of Clinical Nutrition and Metabolism (ESPEN). The validation of the model system based on samples from an IBS cohort was funded by a project grant awarded to Julia Rode by the Faculty of Medicine and Health at Örebro University (Projektbidrag för att främja forskningen 2023 (B) ORU 2022/07087). The in vivo butyrate infusion human trial was funded by the Swedish Research Council (grant number 2017-02694), and the ex vivo human trial by the Swedish Farmers Foundation for Agricultural Research. The funders did not have any involvement in the conduction or interpretation of the project. The graphical abstract was created with Krita v. 4.1.3 and Microsoft PowerPoint v. 16.94.

## Author contributions

R.V., I.R., T.M., R.W., R.B., J.R.—Conceptualization; M.C., S.P., V.C.A., J.R.—Data curation; M.C., S.P., J.R.—Formal analysis; J.R.—Funding acquisition; M.C., R.V., M.S., J.R.—Investigation; M.C., R.V., S.P., V.C.A., R.B., J.R.—Methodology; J.R.—Project administration; R.V., V.C.A., T.M., R.W., R.B., J.R.—Resources; R.V., V.C.A., A.H., R.B., J.R.—Supervision; R.V., J.R.—Validation; M.C., S.P., J.R.—Visualization; M.C., V.C.A., A.H., J.R.—Writing—original draft; All authors—Writing—review and editing.

## Funding

## Competing interests

The authors declare no competing interests.
