## [Transparent Peer Review file · Communications Biology]

Development of a novel humanized gut-brain axis model as a tool toward personalized nutrition

Corresponding Author: Dr Julia Rode

Version 0:

Reviewer comments:

Reviewer #1

(Remarks to the Author)

The manuscript titled, "Development of a novel humanized gut-brain axis model as a tool toward personalized nutrition " is a study focussing on therapeutic or preventive potential of interventions modulating tryptophan metabolism. A major takeaway is that the serosal fluids from the intestinal tissues treated with butyrate or fibre-rich fractions altered the gene and protein levels following oxidative stress.

One of the major concerns in the studies is absence of transepithelial electrical resistance results. It not only reflects on the tissue barrier integrity in the Ussing chamber or transwell membranes but also serves as an indicator of viability of the tissue in ex vivo or in vitro settings. Can authors provide a comment or possibly results for the intestinal tissues in the Ussing chamber or fibroblast in in vitro BBB model used?

Line 134: Were samples from two donors pooled for faecal fermentation? Please specify the exact number as 2) seems confusing to reader.

Line 149: Abbreviations of pooled serosal fluids need full forms.

Line 161-165: Following can be moved to discussion section, Systemic tryptophan, either free or albumin-bound, can be transported through specialized brain microvascular endothelial cells (BMECs) of the BBB by two closely related plasma membrane transporters, large neutral amino acid transporter 1 (LAT1) and 2 (LAT2) 45,46. Commonly abundant in biological barriers, these transporters are also expressed in fibroblasts, making them a well-established in vitro model to study the effects of gut-derived substances on BBB physiology."

Line 313: If it's not statistically significant, it may not need mentioning to be significant. Authors are advised to explain the context of biological relevance with support of results in figure 2.

Line 340: 24 h exposure can be called as relatively extended exposure. Chronic would suggest the condition otherwise.

Line 416: Ussing chambers are quite an open system where it's hard to maintain the sterile conditions. Can authors provide insights into what makes it suitable for use in sterile culture transwells.

Line 418: What is the typical morphology of fibroblasts like?

Reviewer #2

(Remarks to the Author)

Question 1:

When exposing the model to faecal fibre fermentation, why were sample from only healthy patients utilised? Why weren't samples from patient with IBS also investigated?

Question 2:

Relating to the IBS patients, was the diet prior to the sample collection noted, and do you think that this could potentially be a limitation of the study? For example, if a patient with IBS has a flare up prior to the sample collection could this potentially skewer the results as the sample may be more indicative of an inflamed damaged state.

Comment:

Just with figure 2-5 it may be more ideal to change the shade of grey scale used to more obvious difference as it may be a bit difficult for reader to differentiate between treatment groups, especially with some of the columns being small in height.

Comment:

Double check you figure, as you have two figures labelled as figure 4, and make changes within the text to reflect this as well.

Version 1:

Reviewer comments:

Reviewer #1

(Remarks to the Author)

Authors have provided responses to support the information used for integrity of tissues mounted on using chambers, as this was one of the primary concerns. I am happy for the manuscript to be considered for publication based on the improvements made in the manuscript.

Reviewer #2

(Remarks to the Author)

I am happy with the updated version of the paper, thank you for making the amendments to the paper.

Response to reviewers

On behalf of our co-authors, we would like to thank you for reviewing our manuscript titled “*Development of a novel humanized gut-brain axis model as a tool toward personalized nutrition*”. We thoroughly appreciate the reviewers’ insightful and beneficial comments. We have addressed the comments as discussed below, and we have indicated any changes in the manuscript.

Reviewer #1 (Remarks to the Author):

The manuscript titled, "Development of a novel humanized gut-brain axis model as a tool toward personalized nutrition " is a study focussing on therapeutic or preventive potential of interventions modulating tryptophan metabolism. A major takeaway is that the serosal fluids from the intestinal tissues treated with butyrate or fibre-rich fractions altered the gene and protein levels following oxidative stress.

One of the major concerns in the studies is absence of transepithelial electrical resistance results. It not only reflects on the tissue barrier integrity in the Ussing chamber or transwell membranes but also serves as an indicator of viability of the tissue in ex vivo or in vitro settings. Can authors provide a comment or possibly results for the intestinal tissues in the Ussing chamber or fibroblast in in vitro BBB model used?

Thank you for finding merit in our work and raising critical concerns. With regard to the Ussing chamber experiments upon mounting the biopsies and allowing an equilibrium to be reached, potential differences (PD) were assessed and biopsies with a PD>0.5 were excluded and replaced due to uncertain tissue viability. PD, instead of transepithelial electrical resistance (TEER), was used to ensure viability, since it is technically more relevant. Nevertheless, TEER was assessed throughout the experiments and was minimally 9 Ω/cm^2 after reaching an equilibrium and before ex vivo manipulations (T0). We have now included clarifications of this point in lines 593-596.

The fibroblast cells were cultured to a visually determined confluency of 95% but not to a total confluence. While the culture on transwell membranes would have been interesting, it was out of scope for this work, which focused on the alteration of fibroblast gene and protein expression patterns and tryptophan uptake into the cells, but not cross-barrier transport. Therefore, TEER of fibroblast cell cultures was not measured. Viability of fibroblasts in our setting in general was assessed using AlamarBlue and LDH assays. Additionally, for every experiment, the fibroblasts’ viability was confirmed visually, namely that cells were attached to the wells surface and not floating. We have now included clarifications of these points in lines 582-588.

Line 134: Were samples from two donors pooled for faecal fermentation? Please specify the exact number as 2) seems confusing to reader.

Every biopsy was exposed to supernatants of fecal fermentation from its own donor. Hence, biopsies from 12 donors were exposed to fecal fiber fermentations using 12 feces from the respective matched donor. We have now clarified this in lines 114, 115, 119, 121, 134-138.

Line 149: Abbreviations of pooled serosal fluids need full forms.

We have now further explained the abbreviations of serosal fluid pools, lines 155-160. Also, we have ensured that abbreviations are explained in any figure or table legend, where explanations were previously incomplete or missing.

Line 161-165: Following can be moved to discussion section, Systemic tryptophan, either free or albumin-bound, can be transported through specialized brain microvascular endothelial cells (BMECs) of the BBB by two closely related plasma membrane transporters, large neutral amino acid transporter 1 (LAT1) and 2 (LAT2) 45,46. Commonly abundant in biological barriers, these transporters are also expressed in fibroblasts, making them a well-established in vitro model to study the effects of gut-derived substances on BBB physiology.”

We agree with the reviewer, and this has now been moved to the discussion section, lines 577-582.

Line 313: If it's not statistically significant, it may not need mentioning to be significant. Authors are advised to explain the context of biological relevance with support of results in figure 2.

We have now expressed this clearly as “[...] significantly decreased for the HC-CON condition and non-significantly ($p=0.0518$) for the HC-FERFIB condition [...]”, lines 339-340.

As common for gene expression data, we considered alterations to be biologically relevant if downregulated two-times, hence below the 0.5-fold mark, or upregulated two-times, hence above the 2-fold mark. We have now clarified the applied concept of biological relevance further in lines 241-242 and 328-330, and figure legends.

Line 340: 24 h exposure can be called as relatively extended exposure. Chronic would suggest the condition otherwise.

Thank you for this suggestion; the term “chronic exposure” has now been replaced by “relatively extended exposure” throughout the manuscript.

Line 416: Ussing chambers are quite an open system where it's hard to maintain the sterile conditions. Can authors provide insights into what makes it suitable for use in sterile culture transwells.

Indeed, the Ussing chamber experiments were used as an open system where sterile conditions cannot be guaranteed. Therefore, we have assessed microbial growth of serosal fluids (see lines 163-164, and line 266), which was negative. Fibroblasts were cultured with antibiotics (see Supplementary Note “Cell Culture”), and contamination was not detected throughout the duration of experimentation. We have now added a discussion of this point in lines 470-473 and line 478, as well as raised it as potential limitation in lines 596-599.

Line 418: What is the typical morphology of fibroblasts like?

We have now clarified the typical spindle-shaped morphology and adhesive character of fibroblasts also in this place, now lines 475-476 in the revised manuscript. This had also been described in the results section, lines 302-303.

Reviewer #2 (Remarks to the Author):

Question 1:

When exposing the model to faecal fibre fermentation, why were sample from only healthy patients utilised? Why weren't samples from patient with IBS also investigated?

Serosal fluids have been collected from recently conducted human trials that focused on other primary objectives. While the colonic *in vivo* exposure to butyrate was conducted in both healthy participants (HC) and participants with IBS, the *ex vivo* exposure to supernatants from fecal fiber fermentations was solely conducted in HC. Although the assessment of serosal fluids from IBS biopsies *ex vivo* exposed to supernatants from fecal fiber fermentations would have been a strong addition to this work, such samples were not available. However, this is a very valuable question to explore in future research. We have now clarified this in the methods section, line 119, as well as discussed in lines 601-606.

Question 2:

Relating to the IBS patients, was the diet prior to the sample collection noted, and do you think that this could potentially be a limitation of the study? For example, if a patient with IBS has a flare up prior to the sample collection could this potentially skewer the results as the sample may be more indicative of an inflamed damaged state.

All participants received a standardized low-fiber dinner (pasta and tomato sauce) on the evening before the colonoscopy, which they consumed at least 10 hours before the colonoscopy and fasted thereafter.

We have now added this information in the methods section, lines 148-150. Yet, the effect of long-term diet on the results cannot be ruled out, and we have now added this as a limitation in lines 612-613.

All IBS participants reported a Gastrointestinal Symptom Rating Scale for IBS (GSRS-IBS) total score of 3.8 ± 0.8 (mean \pm sd) for the week before the colonoscopy. The GSRS-IBS ranges from '1' (no discomfort) to '7' (very severe discomfort) and a rating of '4' indicates moderate discomfort. No participant rated above '6' (moderately severe discomfort). Our IBS cohort is hence comparable to others. Additionally, on the day of the colonoscopy, participants were asked about changes in their medical condition, and no one reported any. We have now added this information in lines 606-610. To further assess worsening of symptoms prior to the colonoscopy a more long-term measure would have been needed, but such was not available. We have now added this limitation in lines 610-611.

Comment:

Just with figure 2-5 it may be more ideal to change the shade of grey scale used to more obvious difference as it may be a bit difficult for reader to differentiate between treatment groups, especially with some of the columns being small in height.

Thank you for this suggestion; we have now transformed all our bar graphs into boxplots and indicated the results from the different cohorts in more obviously different colors. Experimental controls (RM, G-Krebs) are now presented in black color, HC-CON and HC-BUT are now presented in green color, IBS-CON and IBS-BUT in blue color, and HC-FERCON and HC-FERFIB in purple color. Furthermore, in all figures, the same order is kept. We have also changed the color of the dashed lines indicating various thresholds from blue to grey.

Comment:

Double check you figure, as you have two figures labelled as figure 4, and make changes within the text to reflect this as well.

Thank you for identifying this error. We have now double-checked all our Figures and Supplementary Figures ensuring correct labelling and referencing.

We hope that you find that we have sufficiently addressed the reviewers' comments. Thank you for reviewing and considering this manuscript for publication. We look forward to your response.

Respectfully yours,

Dr Julia Rode MSc PhD